# A-GSTCN: An Augmented Graph Structural–Temporal Convolution Network for Medication Recommendation Based on Electronic Health Records

**DOI:** 10.3390/bioengineering10111241

**Published:** 2023-10-24

**Authors:** Weiqi Yue, Maiqiu Wang, Lei Zhang, Lijuan Zhang, Jie Huang, Jian Wan, Naixue Xiong, Athanasios V. Vasilakos

**Affiliations:** 1School of Electronic and Information Engineering, Zhejiang University of Science and Technology, Hangzhou 310023, China; 222208855044@zust.edu.cn (W.Y.); huangjie@zust.edu.cn (J.H.); wanjian@zust.edu.cn (J.W.); 2Institute of Biochemistry, Zhejiang University of Science and Technology, Hangzhou 310023, China; maiqiu_wang@zust.edu.cn; 3Department of Computer Science and Mathematics, Sul Ross State University, Alpine, TX 79830, USA; xiongnaixue@gmail.com; 4The Center for AI Research (CAIR), University of Agder (UiA), 4879 Grimstad, Norway; thanos.vasilakos@uia.no

**Keywords:** electronic health records, medication recommendation, graph structural-temporal convolutional network, dilated convolution

## Abstract

Medication recommendation based on electronic health records (EHRs) is a significant research direction in the biomedical field, which aims to provide a reasonable prescription for patients according to their historical and current health conditions. However, the existing recommended methods have many limitations in dealing with the structural and temporal characteristics of EHRs. These methods either only consider the current state while ignoring the historical situation, or fail to adequately assess the structural correlations among various medical events. These factors result in poor recommendation quality. To solve this problem, we propose an augmented graph structural–temporal convolutional network (A-GSTCN). Firstly, an augmented graph attention network is used to model the structural features among medical events of patients’ EHRs. Next, the dilated convolution combined with residual connection is applied in the proposed model, which can improve the temporal prediction capability and further reduce the complexity. Moreover, the cache memory module further enhances the model’s learning of the history of EHRs. Finally, the A-GSTCN model is compared with the baselines through experiments, and the efficiency of the A-GSTCN model is verified by Jaccard, F1 and PRAUC. Not only that, the proposed model also reduces the training parameters by an order of magnitude.

## 1. Introduction

Electronic health records (EHRs) are the primary data carrier for personalized medical research and help accelerate the care process and ensure medical quality. With the increasing potential of EHRs for medical applications, a great deal of research has been applied in this field, which includes diagnosis prediction and medication recommendation [1,2,3,4]. As shown in Figure 1, medication recommendation is of great importance because it can simplify the medical process and assist doctors in making accurate prescriptions. The target of medication recommendation is to recommend personalized and precise drugs for patients based on their current diagnosis and their historical health condition, whereas previous medication recommendation research was based on the rules and facts derived from specialists with abundant clinic experience [5,6,7,8]. With the deepening of medical informatization, deep learning models significantly improve the accuracy of medication recommendation tasks and the feasibility for practical application [9,10,11]. Nevertheless, because of the following characteristics, EHRs bring difficulties to medication recommendation tasks:1.**Structural correlation**: A patient’s EHRs can be seen as a combination of a set of diagnoses, procedures and medications, where the diagnoses, procedures and medications can be collectively referred to the medical events. Therefore, the EHRs can be expressed as a combination of multiple medical events, and the occurrences of medical events simultaneously in a medical record are referred to as structural correlations. For example, chemical ulcers are often accompanied by gastric perforation, and chickenpox can cause erysipelas. These phenomena can be considered as structural correlations between diagnostic events and diagnostic events themselves. Similarly, the combination of statins with cardiovascular drugs is more beneficial for recovery from coronary heart disease, and this phenomenon is thought to be structurally correlated with diagnostic events and medication combinations.2.**Temporal dependency**: Chronic diseases, such as stroke, diabetes and high blood pressure, do not recover as quickly as common diseases. On the contrary, chronic diseases are often incurable and require multiple visits. Meanwhile, during the patient’s medical treatment process, different treatments and drugs can be used at different times. The connection of these medical events on a temporal level is referred to as temporal dependency. For the same patient, the EHRs at multiple admissions can be regarded as multiple continuous medical processes, which may have rich temporal characteristics. In addition, different medical events (diagnoses, procedures and medications) may show different temporal dependencies in different patients.

To capture the structural correlation and temporal dependency of the EHRs, a lot of work has been performed in the early research [12,13,14,15]. However, these methods are rule-based or based on simple classifications, resulting in poor learning ability of EHRs. With the gradual popularization of neural-network-based methods, the graph structure is introduced to capture the structural correlation. Some studies [16,17,18] introduce the graph convolutional network (GCN) for structural modeling, which learns the internal correlation between medical events adequately. However, they ignore the temporal dependency of patients’ records, so that the change of EHRs is not restricted and the models cannot recommend medications accurately. Moreover, some models [19,20,21] consider the temporal change of EHRs, but they cannot cope with the medical events with a complex topological structure, which leads to their inability to describe the structural correlation of EHRs.

Therefore, to simultaneously learn the structural correlation and temporal dependency of EHRs, we propose a novel medication recommendation model called augmented graph structural-temporal convolutional network (A-GSTCN). As shown in Figure 2, we use ICD-9 encoding and ATC encoding to standardize the datasets. Moreover, we use an augmented graph attention network (GAT) to learn the structural correlations of EHRs and further utilize dilated convolution combined with residual connection to capture the temporal features.

Our contributions can be summarized as follows:1.We treat EHRs as time-series records with structural correlation and use ICD-9 encoding and ATC encoding to standardize the records in pretraining. Meanwhile, the A-GSTCN model is proposed to realize personalized medication recommendation based on the standardized records, and the model has excellent performance and can be used in specific medical environments.2.In the A-GSTCN model, we construct global structural correlation diagrams for diagnoses and procedures, capturing the structural correlation of EHRs based on these diagrams and augmented GAT. In addition, we learn the temporal dependency of EHRs by dilated convolution combined with residual connection. Furthermore, we employ a cache mechanism to enhance the medication recommendation accuracy of the proposed model.3.The proposed model outperforms the baselines in all evaluation metrics (Jaccard, F1, PRAUC) for the MIMIC-III datasets and ZJ-CVD datasets. Compared to the baselines, the A-GSTCN model has more accurate drug recommendation ability and requires far fewer parameters, which greatly reduces the training time and significantly improves the inference speed.

The subsequent contents are arranged as follows: Section 2 introduces some related work used in the paper, and Section 3 reviews the framework of the A-GSTCN. In Section 4, the A-GSTCN model and the baselines are compared for the MIMIC-III datasets and ZJ-CVD datasets from several angles, and meanwhile, the high efficiency of the proposed model is proved by experiments. Finally, the conclusion and future work are described in Section 5.

## 2. Related Work

Medication recommendation is a significant research direction in the field of medicine, and it can assist doctors to formulate safe and effective prescriptions quickly. Moreover, the existing medication recommendation approaches can be divided into two categories, i.e., model-driven approaches and data-driven approaches.

Early medication recommendation approaches are mainly based on the model-driven approach, which focuses on the rules and the causal relationship among diagnoses, procedures and medication combinations. These model-driven methods require experts in the field of medicine to model medical events in detail based on prior knowledge. Specifically, Chen et al. [22] developed the reasoning templates based on the knowledge patterns to encode the clinical guidelines for chronic heart failure (CHF) management. Ajmi et al. [23] proposed a backward rule-based expert system, which could be used for a headache diagnosis and medication recommendation system. In addition, a backward rule-based expert system [24] is presented, which can be used for a headache diagnosis and medication recommendation system. In addition, medication recommendation can be influenced by many factors, such as different areas of the hospital, different medical habits of doctors and different disease characteristics of patients [12]. Furthermore, medication recommendation rules that rely on experts’ prior knowledge produce a huge amount of work and affect the efficiency of the recommendations [14,15].

With the continuous accumulation of medical records, the data-driven approach has gradually become an important application for medication recommendation. Specifically, Choi et al. [20,21] employed a traditional recursive neural network (RNN) and an attention-based RNN to learn the multiple admission sequence of patients, thereby obtaining the temporal characteristic of EHRs. Pang et al. [25] added medical records to the pretraining module of BERT by using artificial time tokens. In fact, these approaches learn the temporal characteristics of EHRs and further improve the accuracy of medication recommendation. Nevertheless, early data-driven approaches ignore the structural correlation between medical events.

With the continuous deepening of research on medication recommendation, many comprehensive approaches to learn EHR characteristics have appeared. To be specific, Wang et al. [26] proposed an adversarially regularized model for medication recommendation, which could model the temporal information of EHRs and built a key value memory network based on information from historical admissions. Shang et al. [27] proposed a graph augmented memory network named GAMENet, which could integrate the drug–drug interactions and model longitudinal patient records as a query. Methods [28,29] could model the correlation between medical events and learn the structural correlation of EHRs by constructing medical ontology trees. Mao et al. [16] proposed an intelligent medical system that can accurately estimate the lab values and automatically recommend medication combinations based on patients’ incomplete lab tests. Furthermore, the COGNet model [30] introduces a novel copy-or-predict mechanism to generate the set of medicines. While these models have improved the accuracy of medication recommendation compared to previous models, they also have certain limitations, such as difficulty in applying to real environments, high complexity and so on.

For the above reasons, we propose a novel model named A-GSTCN, which can simultaneously model the structural and temporal characteristics of EHRs. Meanwhile, the proposed model can be also used for medication recommendation tasks in practical applications.

## 3. The A-GSTCN Model

The A-GSTCN model is described in three parts. Firstly, the structure of the proposed model and the goal of the medication recommendation tasks are described. Next, the A-GSTCN’ framework is presented. Last but not least, the optimizer and the training algorithm of the proposed model are introduced. For ease of description, the notations used in the A-GSTCN model are shown in Table 1.

### 3.1. Problem Formulation

An efficient medication recommendation model requires high precision of datasets. To improve the availability of the datasets, the EHRs need to be cleaned and standardized. To be specific, the definition of standardized EHRs, the medical event correlation diagram constructed in pretraining and the goal of the medication recommendation tasks are presented as follows.

#### 3.1.1. Standardized EHRs

The pretrained EHRs can be represented as a collection of temporal records as follows: $\;X^{n}$=$\left\{x_{1}^{1},x_{2}^{1},x_{3}^{1},x_{1}^{2},\ldots ,x_{t}^{n}\right\}$, where n∈[1,N],t∈[1,T],N represent the total number of patients and *T* represents the maximum number of one’s visits. To describe the algorithm more clearly, we omit the superscript *n* and introduce the proposed model only by unit patient. Each visit $x_{t}$=$\left\{c_{d}^{t},c_{p}^{t},c_{m}^{t}\right\}$ of a patient contains diagnosis codes, $c_{d}^{t}$, procedure codes, $c_{p}^{t}$, and medication codes, $c_{m}^{t}.$

#### 3.1.2. Medical Events Correlation Diagrams

To obtain the structural correlation between the medical events, we construct a diagnosis graph matrix $G_{d}\in R^{N_{d}\times N_{d}}$ and a procedure graph matrix $G_{p}\in R^{N_{p}\times N_{p}}$ for all the diagnosis events and procedure events, where $N_{d}$ and $N_{p}$ respectively represent the total number of diagnosis events and procedure events in the data set. Moreover, since $G_{d}$ and $G_{p}$ are built in the same way, we use $G_{*}$ to express them. Finally, the positive point-wise mutual information (PPMI) [31] is used to calculate the correlation between medical event *i* and medical event *j* of $G_{*}$. The formula of $G_{*}$ is defined as follows:(1)$$G_{*}\left(i,j\right)=PPMI\left(i,j\right)=max\left(log_{2}\frac{p(i,j)}{p(i)p(j)},0\right),$$
where p(i,j) represents the probability of simultaneous occurrence of the event *i* and event *j*, and p(i) and p(j) represent the probability of event *i* and event *j*, respectively.

#### 3.1.3. Medication Recommendation Tasks

Given a patient’s historical visits $X_{1:t-1}$=$\left[x_{1},x_{2},\ldots ,x_{t-1}\right]$, diagnosis events $c_{d}^{t}$ and procedure events $c_{p}^{t}$ at the *t*th visit, the goal of medication recommendation tasks is to generate a personalized medication combination ${\hat{y}}_{t}$=${\left\{0,1\right\}}^{N_{m}}$ at the *t*th visit based on the patient’s current clinical events $c_{d}^{t}$, $c_{p}^{t}$ and historical visits $X_{1:t-1}$, where $N_{m}$ represents the total number of the medications.

### 3.2. The Framework of A-GSTCN

The A-GSTCN model includes four components: medical entity embedding module, structural correlation enhancement module, temporal dependency progressive module and cache memory enhancement module. Next, the modules presented in Figure 3 and the algorithm processes of the A-GSTCN model will be described as follows.

#### 3.2.1. Medical Entity Embedding Module

The patient’s *t*th visit $x_{t}$ consists of $\left\{c_{d}^{t},c_{p}^{t},c_{m}^{t}\right\}$, where both $c_{d}^{t},c_{p}^{t},c_{m}^{t}$ are multi-hot vectors, so $c_{*}^{t}$ is used to indicate the unified definition. The medical embeddings for $c_{d}^{t},c_{p}^{t}$ are derived separately, and the embedding matrixes $e_{d}^{t}\in R^{|c_{d}^{t}|\times l}$ and $e_{p}^{t}\in R^{|c_{p}^{t}|\times l}$ are obtained by embedding entities, where ${|c}_{d}^{t}|$ and $|c_{p}^{t}|$ represent the total number of diagnosis events and procedure events at the *t*th visit, and *l* represents the characteristic dimensions. Specifically, the embedding formula of $e_{*}^{t}$ ($e_{*}^{t}$ is used for $e_{d}^{t}$ and $e_{p}^{t}$) is shown as follows:(2)$$e_{*}^{t}=W_{*,e}c_{*}^{t}.$$

Here, $W_{*,e}\in R^{N_{*}\times l}$ presents the embedding matrix, and $N_{*}$ is the total number of medical events. Through the medical entity embedded module, the input $x_{t}$=$\left\{c_{d}^{t},c_{p}^{t},c_{m}^{t}\right\}$ is transformed into ${\hat{x}}_{t}$=$\left\{e_{d}^{t},e_{p}^{t},c_{m}^{t}\right\}$.

#### 3.2.2. Structural Correlation Enhancement Module

The function of the structural correlation enhancement module is to make the embedding matrix $e_{*}^{t}$ contain information about other related medical events and obtain a more comprehensive matrix representation. For this reason, we propose an enhanced multi-head graph attention network. Specifically, the medical events correlation diagram $G_{*}$ constructed in pretraining is used as the global weight matrix. For the value $e_{*}^{t}$=$\left\{e_{*,1}^{t},e_{*,2}^{t},\ldots e_{*,|c_{*}^{t}|}^{t}\right\}$, graph transformation is performed for each of its sub-events $e_{*,i}^{t}$ and the hidden layer $h_{*}^{t}$=$\left\{h_{*,1}^{t},h_{*,2}^{t},\ldots h_{*,|c_{*}^{t}|}^{t}\right\}$ is obtained with more structural information. The specific calculation formula [32] of $h_{*,i}^{t}$ can be written as follows:(3)$$h_{*,i}^{t}{=\|}_{k=1}^{K}\sigma \left(\sum_{j\in N_{i}}\alpha_{ij}^{*,t,k}W^{k}e_{*,i}^{t}+b^{k}\right),$$
where ‖ is the concatenation operation; $h_{*,i}^{t}$ represents the sub-event graph transformation; *K* is interpreted as the number of multiple attention; σ represents a nonlinear function; $N_{i}$ can be interpreted as the collection of other sub-events related to the event *i*; $W^{k}$ and $b^{k}$ represent the weight matrix and bias, respectively; $\alpha_{ij}^{*,t,k}$ represents the weight coefficient of attention at the *t*th visit. To be specific, the calculation formula of $\alpha_{ij}^{*,t,k}$ [33] is illustrated as follows:(4)$$\alpha_{ij}^{*,t,k}=\frac{exp(LeakReLU({\vec{a}}^{T}[W{\vec{h}}_{i}||W{\vec{h}}_{j}]))}{\sum_{k\in N_{i}}exp(LeakReLU({\vec{a}}^{T}[W{\vec{h}}_{i}||W{\vec{h}}_{k}]))},$$
where ${\vec{a}}^{T}$ is the feedforward neural network training vector; W represents the weight matrix; ${\vec{h}}_{*}$ can be interpreted as the corresponding eigenvector for events ∗. Inspired by previous research [34], instead of complex pretraining, the medical events correlation diagram $G_{*}$ is applied to calculate the weight of medical events in each visit. Therefore, there is no need to train the specific training parameters, such as ${\vec{a}}^{T}$ and W, and the calculation of $\alpha_{ij}^{*,t,k}$ can be simplified as:(5)$$\alpha_{ij}^{*,t,k}=\frac{exp(G_{*,t}\left(i,j\right))}{\sum_{k\in N_{i}}exp\left(G_{*,t}\left(i,k\right)\right))}.$$

Here, $G_{*,t}\left(i,j\right)$ and $G_{*,t}\left(i,k\right)$ are the correlation between event *i* and event *j*, event *i* and event *k* in the graph matrix $G_{*,t}$, respectively. The graph matrix $G_{*,t}$ is derived from the medical events correlation diagram $G_{*}$ as follows:(6)$$G_{*,t}\left(i,j\right)=\left\{\begin{array}{cc} G_{*}\left(i,j\right), & if\;i,j\in c_{*}^{t};\\ 0, & else. \end{array}\right.$$

Thus, the correlation between medical events are learned from the structure correlation enhancement module, and the more comprehensive diagnosis representation $h_{d}^{t}$ and procedure representation $h_{p}^{t}$ are obtained by Equations (Equation 3), (Equation 5) and (Equation 6). To be specific, ${\hat{x}}_{t}$=$\left\{e_{d}^{t},e_{p}^{t},c_{m}^{t}\right\}$ is transformed to ${\hat{x}}_{t}^{\prime }$=$\left\{h_{d}^{t},h_{p}^{t},c_{m}^{t}\right\}$.

#### 3.2.3. Temporal Dependency Progressive Module

GRU and LSTM are firstly considered to capture the temporal dynamic changes of EHRs, but these models have high memory usage. Thanks to the prior research [35], it is more appropriate to use the method of dilated convolution combined with residual connection to learn the temporal characteristics of EHRs. Specifically, simple convolutional networks can only deal with sequential tasks with relatively small sequence length and perform poorly in long sequential tasks, so they cannot be applied to EHRs with an uncertain number of visits. Therefore, the method of combining dilated convolution with residual connection is considered, and we propose a new approach to capture medical events’ temporal dependency for medication recommendation inspired by references [36,37]. As shown in Figure 4, the dilated convolution contains two more significant parameters: filter and factor. The size of filter is set to 7 and the factor is set to 1. As the hidden layer deepens, the receptive field can cover all values from the length of patients’ visits, and the output results are obtained through the residual connection layer. Specifically, $h_{d}^{t}$ and $h_{p}^{t}$ are trained separately, and the specific inputs of the network are $H_{d}$:$\left[h_{d}^{1},h_{d}^{2},\ldots ,h_{d}^{t}\right]$ and $H_{p}$:$\left[h_{p}^{1},h_{p}^{2},\ldots ,h_{p}^{t}\right]$, which could be expressed by $H_{*}$. After the dilated convolution and residual connection, the output $Q_{*}$:$\left[q_{*}^{1},q_{*}^{2},\ldots ,q_{*}^{t}\right]$ contained temporal characteristics can be obtained as follows:(7)$$Q_{*}=F\left(H_{*},\left\{W_{i}\right\}\right)+H_{*}^{d^{\prime }},$$
where $F(H_{*},\left\{W_{i}\right\})$ is a residual mapping and $W_{i}$ represents the set of parameter matrix. $H_{*}^{d^{\prime }}$ represents the hidden layer results obtained through dilated convolution, and it can be expressed as $H_{*}^{d^{\prime }}$:$\left[F_{*}\left(1\right),F_{*}\left(2\right),\ldots ,F_{*}\left(t\right)\right]$. The $F_{*}\left(t\right)$ in $H_{*}^{d^{\prime }}$ can be derived as follows:(8)$$F_{*}\left(t\right)=\left(H_{*}X_{d^{\prime }}f\right)\left(t\right)=\sum_{i=0}^{k-1}f\left(i\right)\cdot h_{*}^{t-d^{\prime }\cdot i},$$
where $X_{d^{\prime }}$ is the dilation factor and *k* represents the filter size; *t* − $d^{\prime }$·*i* accounts for the direction of the past; f(∗) represents the filter function in the dilated convolution process.

In the temporal dependency progressive module, diagnosis representations $Q_{d}$:$\left[q_{d}^{1},q_{d}^{2},\ldots ,q_{d}^{t}\right]$ and procedure representations $Q_{p}$:$\left[q_{p}^{1},q_{p}^{2},\ldots ,q_{p}^{t}\right]$ are obtained, and they capture rich temporal features by the method of combining dilated convolution with the residual connection. Therefore, ${\hat{x}}_{t}^{\prime }$=$\left\{h_{d}^{t},h_{p}^{t},c_{m}^{t}\right\}$ is transformed into ${\hat{x}}_{t}^{\prime \prime }$=$\left\{q_{d}^{t},q_{p}^{t},c_{m}^{t}\right\}$.

#### 3.2.4. Cache Memory Enhancement Module

The cache memory enhancement module pre-stores the historical records of patients in a dynamic bank with key-value pairs, and it can optimize the current recommendation by comparing the similarity between the current recommendation and the historical records. In addition, the conclusions can be drawn from the research [38] that an effective cache memory enhancement module can improve the model’s learning rate of historical conditions, so the cache memory enhancement module is applied and further divided into four steps:1.Create a query vector of the *t*th visit. To be specific, $q_{d}^{t},q_{p}^{t}$ from the set ${\hat{x}}_{t}^{\prime \prime }$ can be generated a query $q^{t}$ as follows:
(9)$$q^{t}=f\left(q_{d}^{t},q_{p}^{t}\right),$$
where f(∗) represents a transformation function, and this function can connect the diagnosis representation $q_{d}^{t}$ and the procedure representation $q_{p}^{t}$.2.Use the $q^{t}$ and medication representation $c_{m}^{t}$ as dependent variables, and generate the cache records before the *t*th visit in the form of key-value pairs as follows:
(10)$$M^{t}={\left\{q^{t^{\prime }}:c_{m}^{t^{\prime }}\right\}}_{1}^{t-1},$$
where $M^{t}$ is empty when *t* = 1, and $t^{\prime }\in \left(1,t-1\right)$ represents the historical visit before the *t*th visit. $M_{k}^{t}$:$\left[q^{1},q^{2},...,q^{t-1}\right]$ is denoted as the key vector, and $M_{v}^{t}$:$\left[c_{m}^{1},c_{m}^{2},...,c_{m}^{t-1}\right]$ is denoted as the value vector to represent the history cache of the *t*th visit.3.Based on the similarity between the representation vector $q^{t}$ and its historical cache, the attention strategy is applied as follows:
(11)$$o^{t}={\left(M_{v}^{t}\right)}^{T}Softmax\left(M_{k}^{t},q^{t}\right),$$
where the similarity between the key vector matrix $M_{k}^{t}$ and the representation vector $q^{t}$ is first considered. Furthermore, the similarity relationship is obtained by matrix multiplication and activation, and the transposed vector matrix $M_{v}^{t}$ is further multiplied to obtain $o^{t}$.4.Activate $q^{t}$ and $o^{t}$, obtain the multi-label recommended medication combination ${\hat{y}}_{t}$. The formula can be expressed as follows:
(12)$${\hat{y}}_{t}=\sigma \left(q^{t},o^{t}\right),$$
where σ is the activation function.

### 3.3. Optimization

The quality of the medication recommendation model can be explained by the gap between the drug recommendation combination ${\hat{y}}_{t}$ generated by the model and the real drug recommendation combination $y_{t}$. Meanwhile, whether a single drug is recommended can be likened to binary classification, so the task of drug combination recommendation can be further classified into multiple classification problems. In this case, the multi-label margin loss $L_{multi}$ and the binary cross-entropy loss $L_{bce}$ are applied as optimizations, which are combined as model’ optimizer $L_{loss}$ as follows:(13)$$L_{loss}=\alpha *L_{bce}+\left(1-\alpha \right)*L_{multi},$$
(14)$$L_{bce}=-\sum_{t}^{T}\sum_{i}y_{i}^{t}log\sigma \left({\hat{y}}_{i}^{t}\right)+\left(1-y_{i}^{t}\right)log\left(1-\sigma \left({\hat{y}}_{i}^{t}\right)\right),$$
(15)$$L_{multi}=\sum_{t}^{T}\sum_{i}^{|c_{m}|}\sum_{j}^{{\hat{Y}}^{t}}\frac{max(0,1-\left({\hat{y}}_{t}\left[{\hat{Y}}_{j}^{t}\right]-{\hat{y}}_{t}\left[i\right]\right))}{L}.$$

Here, α is the mixture weights; ${\hat{y}}_{i}^{t}$ and ${\hat{y}}_{t}\left[i\right]$ represent the medication *i* in the *t*th visit; ${\hat{y}}^{t}\left[{\hat{Y}}_{j}^{t}\right]$ is the *j*th label indexed by predicted label set ${\hat{Y}}^{t}$.

In summary, Algorithm 1 describes the training algorithm of the A-GSTCN.

**Algorithm 1:** Training algorithm of the A-GSTCN

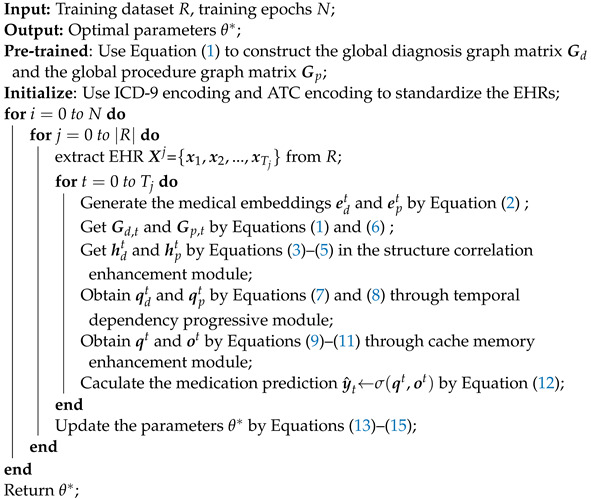



## 4. Experiments

The experiments are divided into three parts. Firstly, preparations of the experimental environment are presented, such as the datasets and the baselines. Secondly, the performance of the A-GSTCN model and baselines is compared in four experiments. Next, a case study is applied for proving the feasibility of the A-GSTCN model in specific medical environments. Finally, through the engineering applications, the A-GSTCN is well applied in the medication recommendation process of a digital hospital.

### 4.1. Experimental Setup

#### 4.1.1. Datasets

The proposed model and the baselines are performed on MIMIC-III and ZJ-CVD datasets, and the relevances of the two datasets are presented as follows:**MIMIC-III** is a sizable single-center database, which includes more than 50,000 cases admitted to intensive care units from 2001 to 2012 and 7870 newborns admitted from 2001 to 2008. To be specific, the MIMIC-III dataset includes medical orders, medications, procedures, diagnoses, and so on. Meanwhile, to improve the dataset availability, the records are generated into a temporal list of diagnosis, procedure and medication codes.**ZJ-CVD** is a Chinese medical dataset collected by our laboratory, which contains the medical records of more than 8000 patients with cerebrovascular disease from the First Hospital of Zhejiang Province, the Fourth Affiliated Hospital Zhejiang University of Medicine and Taizhou Municipal Hospital. Each patient may have multiple hospitalizations, so the number of EHRs in ZJ-CVD datasets exceeds 10,000. To be specific, ZJ-CVD datasets are cleaned and augmented in pretraining and consist of admission diagnosis, hospitalization, discharge medication and some other medical information.

Furthermore, the medical events of the datasets are converted into vector representations according to the ATC and the ICD-9 medical standards. The characteristics of MIMIC-III datasets and ZJ-CVD datasets can be seen in Table 2.

#### 4.1.2. Baselines

The baselines are introduced as follows:**Leap** [39] can predict target event through an attention mechanism by establishing mappings between medical events and tensors.**RETAIN** [21] generates a medication recommendation through building a two-layer RNN with attention model, and this model can consider the influence of temporal factors.**DMNC** [38] strengthens the capturing of temporal characteristics for medical events by establishing a memory enhancement networks.**GAMENet** [27] integrates the drug–drug interactions and model longitudinal patient records as the query, which can capture the temporal dependency of EHRs.**G-Bert** [28] uses the BERT to pretrain the correlations between medical events in EHRs and constructs an ontological tree for medication recommendation.

#### 4.1.3. Metrics

Jaccard Similarity Score (Jaccard), Precision–Recall AUC (PRAUC) and Average F1 (F1) are used as the scoring functions in the experiments. Next, the scoring functions are explained separately.

The caculation formula of Jaccard can be described as follows:(16)$$Jaccard=\frac{1}{\sum_{k}^{N}\sum_{t}^{T_{k}}1}\sum_{k}^{N}\sum_{t}^{T_{k}}\frac{|Y_{t}^{\left(k\right)}⋂{\hat{Y}}_{t}^{\left(k\right)}|}{|Y_{t}^{\left(k\right)}⋃{\hat{Y}}_{t}^{\left(k\right)}|},$$
where *N* is the total number of patients, and $T^{k}$ represents the max visits of the *k*th patient.

PRAUC is calculated by the trapezoidal integral for the area under the PR curve, and this scoring function is used for the datasets with imbalanced positive and negative sample numbers.

The F1 score can transform the multi classification problem into *n* bipartitions. Meanwhile, it calculates the average score of the bipartition to obtain the final evaluation index, whose caculation formula can be written below:(17)$$Avg\left(P_{t}^{\left(k\right)}\right)=\frac{|Y_{t}^{\left(k\right)}⋂{\hat{Y}}_{t}^{\left(k\right)}|}{|Y_{t}^{\left(k\right)}|},Avg\left(R_{t}^{\left(k\right)}\right)=\frac{|Y_{t}^{\left(k\right)}⋂{\hat{Y}}_{t}^{\left(k\right)}|}{|{\hat{Y}}_{t}^{\left(k\right)}|},$$
(18)$$F1=\frac{1}{\sum_{k}^{N}\sum_{t}^{T_{k}}1}\sum_{k}^{N}\sum_{t}^{T_{k}}\frac{2\times Avg\left(P_{t}^{\left(k\right)}\right)\times Avg\left(R_{t}^{\left(k\right)}\right)}{Avg\left(P_{t}^{\left(k\right)}\right)+Avg\left(R_{t}^{\left(k\right)}\right)},$$
where *t* represents *t*th visit, and *k* can be interpreted as the *k*th patient in the test set.

### 4.2. Experimental Results

The effectiveness of the A-GSTCN model is demonstrated by four comparative experiments. Specifically, the A-GSTCN model is compared with the baselines on Jaccard, F1 and PRAUC in the first experiment. In the second part, the validity of each module of A-GSTCN is verified. Next, the third part compares the drug recommendation performance of the model on different recommended frequency drugs. Finally, the last experiment compares the drug recommendation performance of the model for patients with different visits.

#### 4.2.1. Recommendation Performance

Table 3 indicates the comparisons of Jaccard, PRAUC and F1 between the proposed model and the baselines on MIMIC-III and ZJ-CVD datasets. Among them, it is obviously observed that the A-GSTCN model obtains the best recommendation performance under all evaluation metrics, which can prove the effectiveness of the A-GSTCN in medication recommendation. To be specific, compared with the previous best method (G-Bert), the A-GSTCN model improves 1.78%, 1.24% and 1.86% in Jaccard, PRAUC and F1 score, respectively, for the MIMIC-III dataset. In a similar way, the A-GSTCN model increases 2.76%, 8.37% and 2.67% in Jaccard, PRAUC and F1 score, respectively, for the ZJ-CVD dataset. Moreover, the average recommended number of medications for A-GSTCN for the MIMIC-III datasets and ZJ-CVD datasets are 15.34 and 13.22, which have the smallest gap with the real value of 14.61 and 12.89. Futhermore, compared with the baseline methods, the most significant feature of the A-GSTCN model is the correlation diagrams for pretrained medical events and the dilated convolution applied in the temporal dependency progressive module. These features lead to fewer parameters in the A-GSTCN model, which effectively decreases the memory occupancy rate and cache training pressure.

#### 4.2.2. Module Validity

To further prove the effectiveness of the structure correlation enhancement module, the temporal dependency progressive module and the cache memory enhancement module, the A-GSTCN model is compared with its variants.

Variant types of the A-GSTCN model in Figure 5a,b are shown below:**A-GSTCN**: the proposed model.**A-GSTCN (w/o GAT)**: removes the structure correlation enhancement module of the A-GSTCN model.**GAT + GRU**: changes the temporal dependency progressive module into the GRU model for the A-GSTCN model.**A-GSTCN (w/o ME)**: removes the cache memory enhancement module of the A-GSTCN model.

By comparing the performance of the A-GSTCN and the A-GSTCN (w/o GAT) in Figure 5a,b, it indicates that the performance of each metric has a significant decrease when the structural correlation enhancement module is removed. Specifically, Jaccard and F1 score decrease by nearly 8% and 6%, and PRAUC decreases by nearly 16% for the ZJ-CVD datasets. The reductions in Jaccard, F1 score and PRAUC for the MIMIC-III datasets are more prominent. Therefore, it can be concluded that the structural correlation enhancement module behaves excellently in structural modeling and can adequately capture the structural characteristics of medical entities of EHRs.

Through the comparative experiments of the A-GSTCN and the GAT + GRU in Figure 5a,b, it apparently shows that Jaccard, F1 Score and PRAUC for the GAT + GRU decrease by nearly 2% compared with the A-GSTCN for the MIMIC-III datasets, and these metrics decline by nearly 2%, 2%, 6.16% for the GAT + GRU compared with the A-GSTCN model for the ZJ-CVD datasets. Therefore, the conclusions can be drawn from the significant reduction in metrics: use dilated convolution instead of GRU can reduce the amount of parameters used while maintaining model performance in A-GSTCN.

Compared with the proposed model, the Jaccard, PRAUC and F1 score for the A-GSTCN (w/o ME) decline by 1.17%, 1.5% and 1.08%, respectively, for the MIMIC-III datasets. These metrics decline by nearly 4.86%, 15.79% and 7.59% for the ZJ-CVD datasets also. Meanwhile, it is obviously observed that the performance gap between the A-GSTCN and the A-GSTCN (w/o ME) for the ZJ-CVD datasets is larger than that for the MIMIC-III datasets because of the relatively short number of patient visits in the ZJ-CVD datasets. In summary, the cache memory enhancement module can cooperate with the temporal dependency progressive module to fully preserve the temporal features of EHRs, thus improving the accuracy of medication recommendation.

#### 4.2.3. Comparison for Different Recommended Frequency Drugs

Some drugs have a high recommended frequency, and others may be used less often. The A-GSTCN model can decrease the impact of data imbalance by applying the global structural correlation diagrams for diagnoses and procedures and adding a caching mechanism. Specifically, Figure 6a,b count the number of medications in different recommended frequencies in the MIMIC-III and ZJ-CVD datasets, and it can be seen that 58 of the 145 medication types appear less than 100 times, while nearly 40 types are recommended more than 1000 times in the MIMIC-III datasets. In the ZJ-CVD datasets, 133 of the 453 medication types are recommended less than 100 times, while nearly 40 types occur more than 1000 times. Figure 6c,d calculate the average F1 score of medication recommendation results in different recommended frequencies, and it indicate that the A-GSTCN model significantly improves the recommended accuracy of less frequent medications based on its global structural correlation diagrams and caching mechanism.

#### 4.2.4. Comparison for Patients with Different Visits

As shown in Table 2, the max visits of patients in the MIMIC-III and ZJ-CVD datasets are 29 and 4, respectively. Logically speaking, different numbers of admissions of patients also affect the accuracy of medication recommendation. To be specific, Figure 7a,b indicate the comparisons of average F1 score between the A-GSTCN model and baselines with different temporal lengths of EHRs in the MIMIC-III and ZJ-CVD datasets, and it can be found that the A-GSTCN model is superior to the baselines over most of the temporal horizon, especially for long sequences. Meanwhile, it can be apparently observed that the A-GSTCN model also has a significant learning ability in short visit sequences and recommends more precise medication combination for patients than the baseline models. These results prove that the A-GSTCN model has efficient modeling ability for long temporal dependency.

### 4.3. Case Study

To clearly clarify the effectiveness of the A-GSTCN model in the task of drug recommendation, we further compare the drug recommendation results of the model through two specific cases.

The first special case is tested for the MIMIC-III dataset. This case selects a patient’s EHRs of four temporal admissions in the test set, and the patient has various symptoms, such as gout, depression and heart disease. As can be seen in Table 4, the correct recommended combination of drugs for the patient is 15 drugs, and the A-GSTCN model performed best in this case, recommending the right 14 drugs. In contrast, the model with the best recommendations in the baselines is G-Bert, which recommends 13 drugs correctly and misses 2. Other models in baselines are less effective. Moreover, it can be seen that none of the models successfully hit the drug “Anxiolytics”, and this is where subsequent models need to improve.

Similar to Table 4, Table 5 represents a recommended result of a patient who accesses a total of three visits from the ZJ-CVD datasets, and this patient suffered from stroke, diabetes and high blood pressure. In addition, compared with the MIMIC-III datasets, this typical case evidently reflects the recommendation ability of the A-GSTCN model in medication recommendation. Specifically, it obviously shows that the actual number of recommended drugs in the patient’s last visit is eight. Meanwhile, the DMNC model, GAMENet model and G-Bert model perform best among all baselines, but they only recommend five drugs correctly. In contrast, the A-GSTCN model correctly recommends seven drugs and misses only one drug. Furthermore, the missed drug “Rabeprazole Sodium Enteric-coated Capsules” from the A-GSTCN model is also lost in all baseline models due to the low utilization rate of this drug.

Compared with other baseline models, the A-GSTCN model achieves the best medication recommendation effect in both cases, which fully proves that A-GSTCN model can better learn the structural correlation and temporal dependency of EHRs.

### 4.4. Engineering Applications

Medical service informatization is the development trend of Internet medical treatment in the digital age. With the rapid development of information technology, more and more hospitals are accelerating the overall construction of hospital information systems (HISs) to improve the service level and core competitiveness of hospitals. As a new application of the Internet in the medical industry, the digital hospital is an important form of medical service informatization [40]. Since the requirements to ensure the universality and accuracy of medical services, most of the current research focuses on applying deep learning models to learn the structural–temporal characteristics of medical data and then apply these models to medical services, such as medication recommendation, diagnostic prediction, treatment guidance, etc. [41]. Among them, medication recommendation is one of the key issues in the research on the digital hospital. Figure 8 presents the link of medication recommendation in Internet medical treatment.

However, the structural–temporal characteristics of medical records have a great influence on the accuracy of medication recommendation, which directly affects the applicability of the final recommended prescriptions. In this regard, the priority is to produce more accurate deep learning models that can intelligently generate recommended medications. Therefore, as shown in Figure 9, the data-driven approach can be used to collect medical data from patients in cooperative hospitals and clinics for integration into the A-GSTCN model. To be specific, firstly, real medical records are imported into the A-GSTCN model. Then, the structural correlation enhancement module and the temporal dependency progressive module are employed to learn the structural–temporal characteristics of the data, respectively, so as to optimize the recommendation performance of the model and recommend more accurate prescriptions.

## 5. Conclusions and Future Work

In this article, we propose a novel medication recommendation model that can effectively learn the structural correlation and temporal dependency of EHRs. To be specific, we establish the global correlation diagrams for medical events and apply an augmented GAT to capture the structural correlation. Next, dilated convolution combined with residual connection are used to capture temporal features on the premise of greatly reducing training parameters. Meanwhile, the caching mechanism is introduced to improve the medication recommendation accuracy. Finally, through comparative experiments, case studies and engineering applications, it proves that the proposed model has higher medication recommendation accuracy and better landing possibility compared to the previous models.

In light of the current situation, the EHRs introduce a significant amount of uncertainty into medication recommendations due to the lack of information, imprecise information and contradictory nature. Therefore, it is essential to explore the characteristics of other important influencing factors in EHRs, such as inspection indicators and operation status. Meanwhile, as we continuously collect and integrate the EHRs, it is important to consider the introduction of pretrained models like BERT, GPT and other large language models to enhance the performance of the recommendation model. Furthermore, the application of EHRs needs to be expanded; in addition to medication recommendation, it also can be further applied to disease prediction, disease prevention and other issues. Finally, in the process of medication recommendation, it is significant to consider the safety of medication recommendation, and we need to further consider adding drug–drug interactions (DDIs) to ensure the safety of recommended drugs.

## Figures and Tables

**Figure 1 bioengineering-10-01241-f001:**
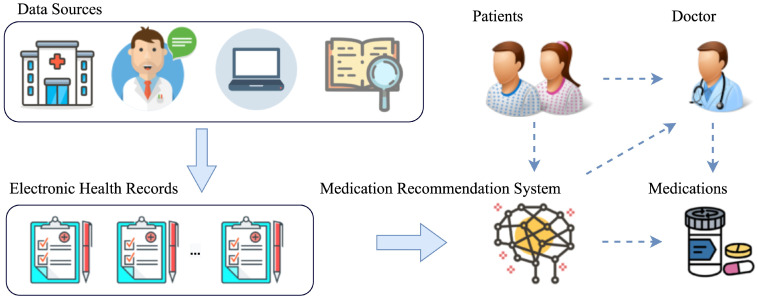
The application of medication recommendation system in a medical scenario. The medication recommendation system learns the collected EHRs in advance and establishes the model to facilitate follow-up patients’ medical treatment and discharge with drugs.

**Figure 2 bioengineering-10-01241-f002:**
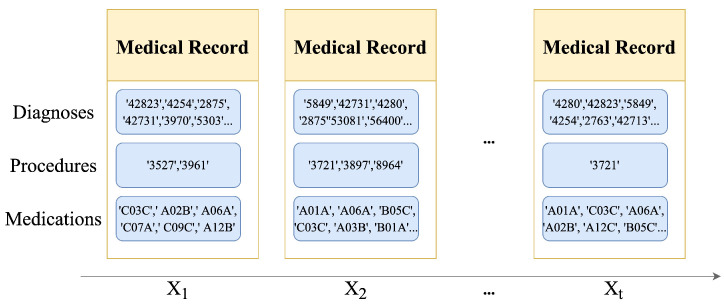
A standardized sample of EHRs. ICD-9 encoding and ATC encoding are used to standardize the EHRs.

**Figure 3 bioengineering-10-01241-f003:**
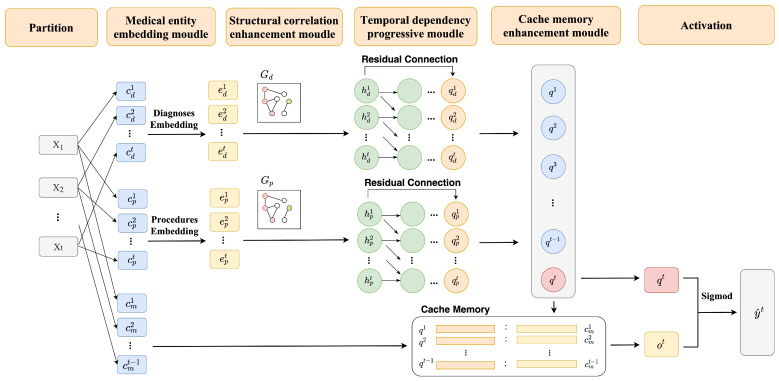
The training process of A-GSTCN model. Each visit $x_{t}$=$\left\{c_{d}^{t},c_{p}^{t},c_{m}^{t}\right\}$ of a patient contains diagnosis codes, $c_{d}^{t}$, procedure codes, $c_{p}^{t}$, and medication codes, $c_{m}^{t}$. Among them, $c_{d}^{t},c_{p}^{t}$ are used in the medical entity embedding module to output the hidden embedding $e_{d}^{t},e_{p}^{t}$ with Equation (Equation 1). Then, structural correlation enhancement module generates $h_{d}^{t},h_{p}^{t}$ by accepting $e_{d}^{t},e_{p}^{t},G_{d}$ and $G_{d}$ described in Equations (Equation 1) and (Equation 3)–(Equation 6). Next, $h_{d}^{t},h_{p}^{t}$ are input into the temporal dependency progressive module to output $\left[q^{1},q^{2},\ldots ,q^{t}\right]$ using the dilated convolution combined with residual connection by Equations (Equation 7) and (Equation 8). After that, the output $o^{t}$ is generated by integrating the key-value pairs stored in cache memory using Equations (Equation 9)–(Equation 11). In the end, query $q^{t}$ and output $o^{t}$ are activated by Equation (Equation 12) for medication recommendation.

**Figure 4 bioengineering-10-01241-f004:**
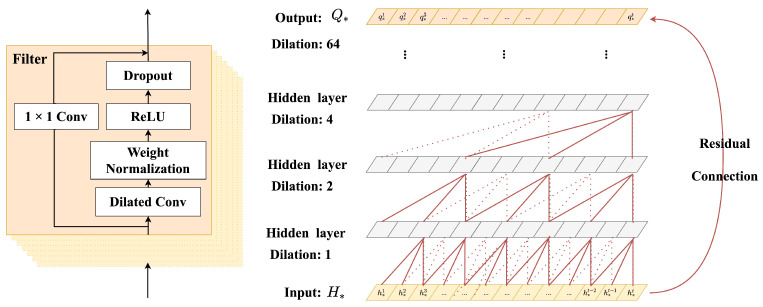
The structure of the temporal dependency progressive module. Both residual and parameterized skip connections are used throughout this module.

**Figure 5 bioengineering-10-01241-f005:**
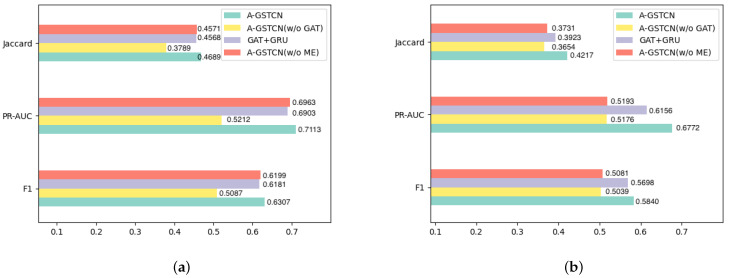
(**a**,**b**) are the performance comparisons (Jaccard, PRAUC and F1 score) between different variants of proposed methods on MIMIC-III and ZJ-CVD datasets.

**Figure 6 bioengineering-10-01241-f006:**
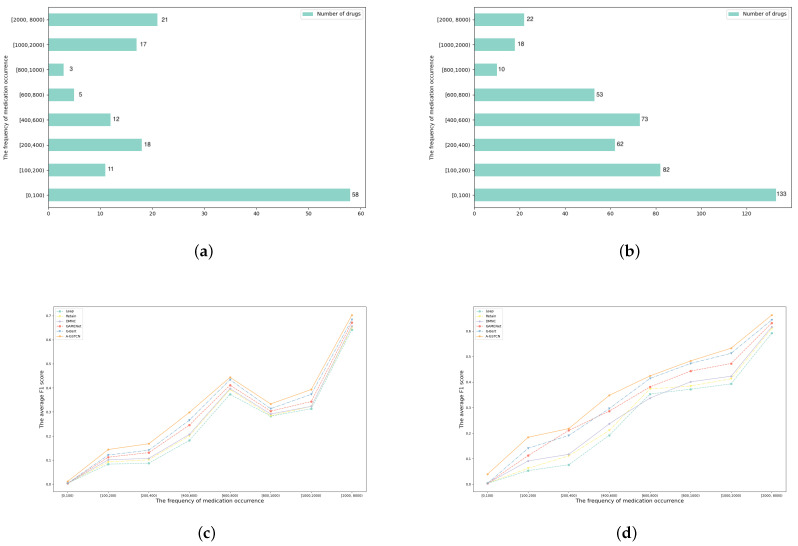
(**a**,**b**) are the total number of medications in different frequency ranges in MIMIC-III and ZJ-CVD datasets; (**c**,**d**) are the comparisons of average F1 score between the A-GSTCN model and baselines in different frequency ranges in MIMIC-III and ZJ-CVD datasets.

**Figure 7 bioengineering-10-01241-f007:**
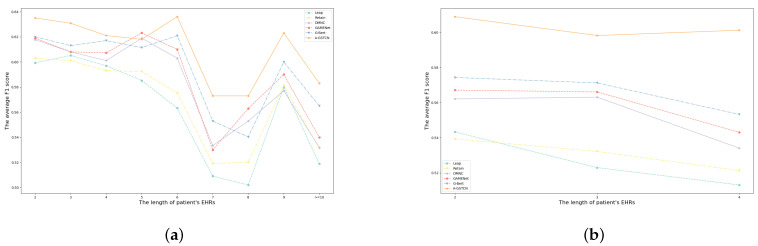
(**a**,**b**) are the comparisons of average F1 score between the A-GSTCN model and baselines with different temporal length of EHRs in MIMIC-III and ZJ-CVD datasets.

**Figure 8 bioengineering-10-01241-f008:**
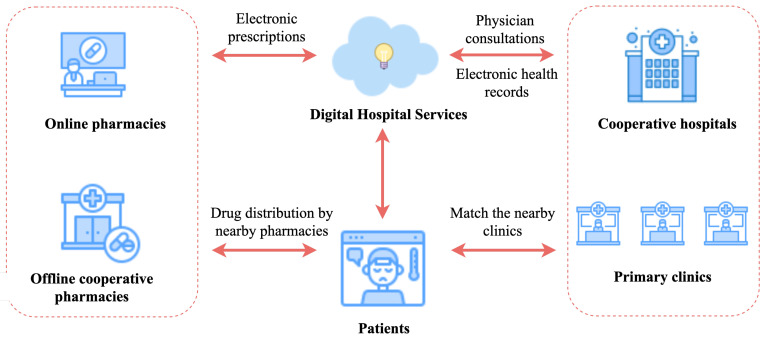
Medication recommendation process in Internet medical treatment.

**Figure 9 bioengineering-10-01241-f009:**
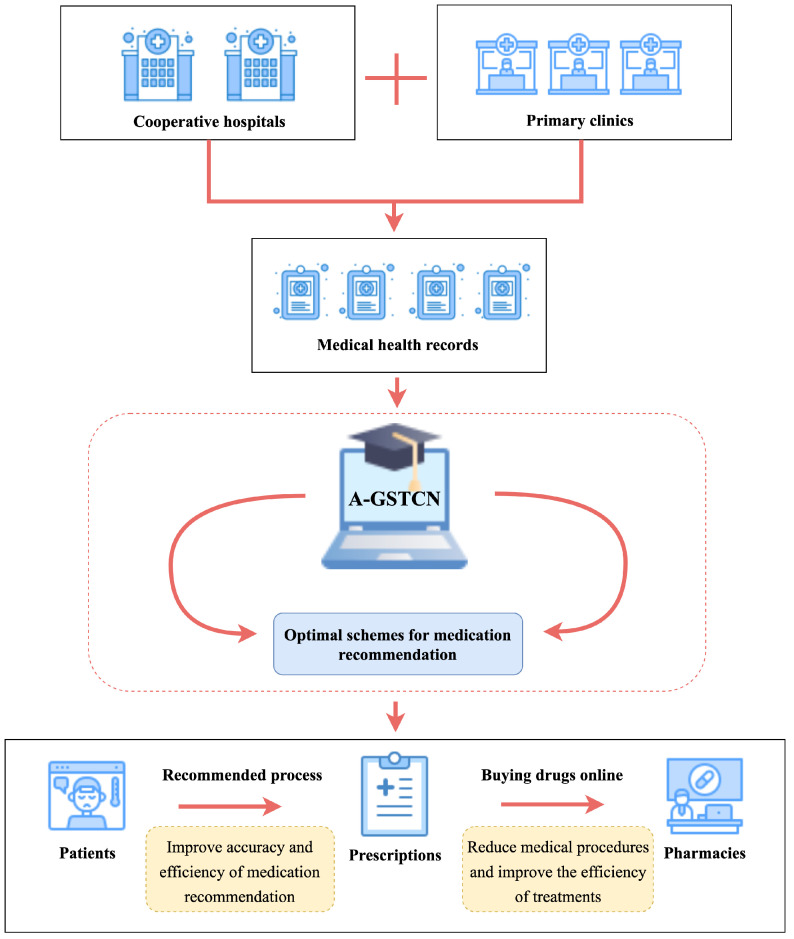
An application diagram of the A-GSTCN in medication recommendation.

**Table 1 bioengineering-10-01241-t001:** Notations used in the A-GSTCN model.

Notation	Description
$X^{n}$	the representation of the pretrained EHRs
$X_{1:t-1}$	the historical visit representation of *t*th visit
$x_{t}$	the representation of *t*th visit
$c_{d}^{t},c_{p}^{t},c_{m}^{t}$	the diagnosis codes, procedure codes and medication codes of *t*th visit
$G_{d},G_{p}$	the global structural correlation diagrams for diagnoses and procedures
$G_{*}$	the representation of $G_{d}$ and $G_{p}$
$N_{d},N_{p},N_{m}$	the total number of diagnoses, procedures and medications
$e_{d}^{t},e_{p}^{t}$	the representations for diagnoses and procedures through medical entity embedding module
$e_{*}^{t}$	the representation of $e_{d}^{t}$ and $e_{p}^{t}$
${\hat{x}}_{t}$	the outputs through medical entity embedding module
$h_{d}^{t},h_{p}^{t}$	the representations for diagnoses and procedures through structural correlation enhancement module
$h_{*}^{t}$	the representation of $h_{d}^{t}$ and $h_{p}^{t}$
${\hat{x}}_{t}^{{}^{\prime }}$	the outputs through structural correlation enhancement module
$H_{d},H_{p}$	the representation of $\left[h_{d}^{1},h_{d}^{2},...,h_{d}^{t}\right]$ and $\left[h_{p}^{1},h_{p}^{2},...,h_{p}^{t}\right]$
$H_{*}$	the representation of $H_{d}$ and $H_{p}$
$H_{*}^{d^{\prime }}$	the representation of hidden-layer results obtained through dilated convolution
$q_{d}^{t},q_{p}^{t}$	the representations for diagnoses and procedures through temporal dependency progressive module
$Q_{d},Q_{p}$	the representation of $\left[q_{d}^{1},q_{d}^{2},...,q_{d}^{t}\right]$ and $\left[q_{p}^{1},q_{p}^{2},...,q_{p}^{t}\right]$
$Q_{*}$	the representation of $Q_{d}$ and $Q_{p}$
${\hat{x}}_{t}^{{}^{\prime \prime }}$	the outputs through temporal dependency progressive module
$q^{t}$	the query vector of the cache memory
$M_{k}^{t},M_{v}^{t}$	the *t*th visit of key vector and the *t*th visit of value vector in cache memory
$M^{t}$	the cache records before the *t*th visit in the form of key-value pairs
$o^{t}$	the memory outputs through the cache memory enhancement module
${\hat{y}}_{t}$	the multi-label medication recommendation of *t*th visit
$\hat{Y}$	the recommended medication set
Y	the ground truth of the medication set

**Table 2 bioengineering-10-01241-t002:** The characteristics of MIMIC-III and ZJ-CVD datasets.

	MIMIC-III	ZJ-CVD
patients	35,886	8315
- single-visit	28,936	6835
- multiple-visit	6950	1480
clinical events	3529	1237
- diagnosis	1958	552
- procedure	1426	232
- medication	145	453
max visits	29	4
average visits	2.36	1.32
average number of diagnosis	10.51	4.15
average number of procedure	3.84	1.20
average number of medication	8.80	6.20

**Table 3 bioengineering-10-01241-t003:** Medication recommendation performance between the A-GSTCN model and baselines on MIMIC-III and ZJ-CVD datasets. In addition, the gold average number of medicines on the test set is 14.61 and 12.89 for the MIMIC-III datasets and ZJ-CVD datasets, respectively.

	MIMIC-III					ZJ-CVD				
**Methods**	**Jaccard**	**PRAUC**	**F1**	**Avg #** **of Med**	**Parameters**	**Jaccard**	**PRAUC**	**F1**	**Avg #** **of Med**	**Parameters**
Leap [39]	0.3844	0.5501	0.5410	13.42	436,884	0.3738	0.5223	0.5187	11.47	303,286
RETAIN [21]	0.4168	0.6620	0.5781	16.68	289,490	0.3769	0.5261	0.5211	12.08	230,254
DMNC [38]	0.4343	0.6856	0.5934	20.00	527,979	0.3803	0.5399	0.5291	16.12	444,143
GAMENet [27]	0.4489	0.6911	0.6053	13.89	452,434	0.3811	0.5418	0.5369	10.71	323,147
G-Bert [28]	0.4511	0.6989	0.6121	16.11	2,411,138	0.3941	0.5935	0.5573	14.41	1,616,783
**A-GSTCN**	**0.4689**	**0.7113**	**0.6307**	**15.34**	**97,626**	**0.4217**	**0.6772**	**0.5840**	**13.22**	**73,424**

**Table 4 bioengineering-10-01241-t004:** A specific case selects a patient’s EHRs of four temporal admissions from the MIMIC-III datasets; “unseen” indicates the drugs that do not appear in the actual recommendation results, and “missed” refers to the drugs that should be recommended in the actual situation but are not recommended.

Methods	Recommended Medication Combination (the Last Visit)
Leap	8 correct + 2 unseen + 7 missed (Antigout, Anxiolytics, Cardiac glycosides, …)
RETAIN	10 correct + 4 unseen + 5 missed (Antigout, Anxiolytics, Potassium, …)
DMNC	11 correct + 6 unseen + 4 missed (Anxiolytics, Cardiac glycosides, Potassium, …)
GAMENet	12 correct + 2 unseen + 3 missed (Antigout, Anxiolytics, Dopaminergic agents)
G-Bert	13 correct + 4unseen + 2 missed (Anxiolytics, Potassium)
**A-GSTCN**	**14 correct + 3 unseen + 1 missed (Anxiolytics)**

**Table 5 bioengineering-10-01241-t005:** A specific case of a patient who accesses a total of three visits from ZJ-CVD datasets, and this patient suffered from stroke, diabetes and high blood pressure. Missing drugs include Rabeprazole Sodium Enteric-coated Capsules (RSEC), Betahistine mesilate Tablets (BMT), Trimetazidine Hydrochloride Tablets (THT), Perindopril And Indapamide Tablets (PAIT) and Aspirin Enteric-Coated Sustained Release Tablets (AESRT). For convenience, corresponding abbreviations are used below.

Methods	Recommended Medication Combination (the Last Visit)
Leap	4 correct + 4 unseen + 4 missed (RSEC, BMT, THT, PAIT)
RETAIN	4 correct + 2 unseen + 4 missed (RSEC, BMT, AESRT, PAIT)
DMNC	5 correct + 2 unseen + 3 missed (RSEC, BMT, AESRT)
GAMENet	5 correct + 3 unseen + 3 missed (RSEC, THT, PAIT)
G-Bert	5 correct + 2 unseen + 3 missed (RSEC, BMT, THT)
**A-GSTCN**	**7 correct + 1 unseen + 1 missed (RSEC)**

## Data Availability

Not applicable.

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
