# Peer review of "A-GSTCN: An Augmented Graph Structural–Temporal Convolution Network for Medication Recommendation Based on Electronic Health Records"

_bioengineering, 2023, doi:10.3390/bioengineering10111241_

Round 1
Reviewer 1 Report
-The paper although interesting, is related to a topic that has already been thoroughly studied and investigated. Least some indications should be provided in the text before acceptance.
-What is the major contribution of the work? Are the review, classification, and discussion of recent works in the area? An update on related work is needed.
-Little is described or explained about figures and tables in the text. The authors should keep in mind that this manuscript is a survey, where the authors must target instructing the readers and clearly describing concepts and arguments.
-The system model explanation could be better explored since there are many variables used without proper contextualization. Suggestion: A table with descriptions of the notations would help in understanding.
-The results do not present a statistical evaluation that is fundamental to the statements made in the text. In addition, the results are not clearly described to confirm the validation of the proposal. It is necessary to rewrite the results, presenting a deeper discussion of the objectives achieved.
-The conclusions must be increased.
Moderate editing of English language required
Reviewer 2 Report
The paper proposes an Augmented Graph Structural-Temporal Convolutional Network (A-GSTCN). It first uses an augmented graph attention network to model the structural features of medical events in a patient's electronic medical record. Then, dilation convolution combined with residual concatenation is applied to the A-GSTCN, which improves the temporal prediction capability and further reduces the complexity. In addition, the paper introduces a cache memory module to further enhance the model's ability to learn the history of the EHR.
In general, the topic is interesting and the problem is defined well. However, there are still the following problems:
1) I am a bit confused by the introduction, and I suggest the authors to add illustrations to elaborate the ideas of structural correlation and temporal dependency mentioned in the paper.
2) The related work also lacks a comparative analysis with state-of-the-art methods.
3) In the experimental part, there are few baseline methods and no comparison with the state-of-the-art methods. Therefore, it is not sufficient to prove the effectiveness of the proposed method in this paper.
4) In the process of drug recommendation, it is suggested to consider the safety of medication recommendation, such as adding drug-drug interactions (DDIs) to ensure the safety of recommended drugs.
There are some minor errors in the paper, and the authors are advised to make corrections. For example, in line 299, datasetsare -> datasets are.
Reviewer 3 Report
In line 44 and line 45, ‘a lot of work’ needs to be clarified more. Along the same line, it is imperative to state your research questions more concretely, possible be in a separate section.
In Line 55, why is the proposed approach ‘novel’?
References are needed in lines 97 and 98.
In line 211, more clarification is needed for ‘relatively poor’. Along the same line, more details are needed for the process of ZJ-CVD datasets collection from line 292 to 298. Specifically, it has been unclear how the 10,000 standardized EHRs are generated.
With reference to lines 324 and 325, have you dealt with any imbalanced positive and negative sample numbers?
In line 380, does ‘evidently’ means that the result has been surprising?
From lines 428 to 431, clarification is needed for the selection of two specific cases. Perhaps, linking with the research questions could enhance the article to a considerable degree.
In line 297, is Python used for data cleaning and standardization? It could add an enhanced value to the project, if the codes are made available in any repository such as GitHub.
In line 389, what has exactly meant by memory pressure? Perhaps more clarification could be useful. The research limitations need to be mentioned in a more subtle manner. It could be datasets limitation or any operational or algorithmic issues. In line 487, some of the influencing factors need to be mentioned. Along the same line, it needs to be mentioned about some of the processes involved in digital hospital. Perhaps an indication with a reference or two could be helpful for the readers.
Round 2
Reviewer 1 Report
The authors responded appropriately to the comments.
Minor editing of English language required